# Reviewing the 95 sociotechnical barriers to the decarbonization of buildings

Erin Heinz [1] ✉, Benjamin K. Sovacool [1,2,3,4], Thomas Kwan [5] &
Vincent Petit [5]

Building decarbonization is critical to meet climate goals, given the embodied energy, and material resources needed to construct buildings, their operational energy use, and the need for new facilities for growing populations in addition to retrofitting inefficient existing structures. Here, we examine 95 sociotechnical barriers that inhibit progress on decarbonization of the building sector that can be categorized by economic, political, social, behavioral, and technical dimensions. We find that economic barriers are the most prevalent globally, followed by political barriers. The building sector's slowness in adopting decarbonization strategies can be explained by its distinct complexity and differences among barriers by building stage, stakeholders of interest, geography, and even the forum of discussion. We also show how the use of the "carbon lock-in" framework validates the intersectional nature of barriers and infers that policy interventions to advance building decarbonization must be dynamic, local, and engage multiple dimensions simultaneously to drive change.

Building decarbonization is crucial for meeting long-term climate goals. Commercial and residential buildings account for 36% of energy use in the United States due to heating, cooling, lighting, and other electricity needs, including 75% of all electricity use and 35% of the nation's carbon emissions[1–3]. Moreover, the scale and scope of building decarbonization challenges remain substantial. In the European Union, where buildings contribute 36% of energy related greenhouse gas emissions, 35 million building units will need renovations by 2030 requiring € 275 billion of additional investment per year[4]. The global investment required to retrofit buildings to align with existing climate and energy policy goals is estimated to be about $ 500 billion United States dollars (USD) annually[5]. However, without improvements and policy interventions, the energy used in buildings can increase by 46–73% in 2050 from the 128 exajoules (EJ) used in 2019, driven by population growth, greater diffusion and use of energy-consuming devices, and increasing living standards in low and middle-income countries[6,7]. For reasons such as these, one study warns that "building operations will be the most critical step in completing the "last mile" of global carbon neutrality."[8]

The building sector is a high impact space for decarbonization because production of a building relies on an entire global supply chain of carbon-intensive materials, (i.e. steel and concrete) and must reconcile the vested interests of a wide network of stakeholders. Decarbonizing the construction sector is a challenge because it requires countries to either retrofit a large stock of existing buildings to meet efficiency standards, or construct new buildings that must bypass conventional, carbon-intensive technologies in favor of low-carbon alternatives[9]. Although energy efficiency has improved, leading to a 12% decrease in per-square-foot energy use in the United States since 2012, overall energy consumption is expected to rise due to population growth and increasing lifestyle demands[10]. The most energy intensive buildings are used for food service, groceries, and healthcare, while the least intensive are used for warehousing and religious services[11].

[1]Institute for Global Sustainability, Boston University, Boston, USA. [2]Department of Earth and Environment, Boston University, Boston, USA. [3]Department of Business Development and Technology, Aarhus University, Aarhus, Denmark. [4]Bennett Institute for Innovation and Policy Acceleration, University of Sussex Business School, Falmer, East Sussex, United Kingdom. [5]Sustainability Research Institute, Schneider Electric TM, Boston, USA. ✉e-mail: eheinz@bu.edu

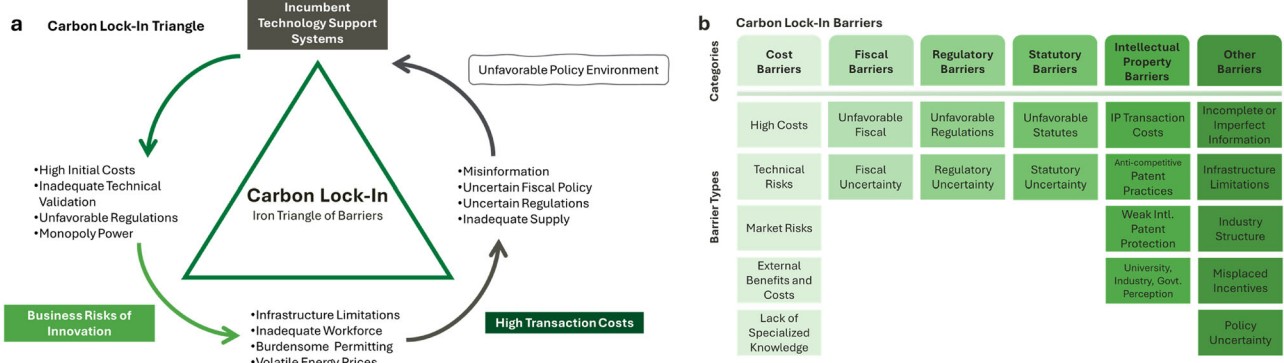

**Fig. 1 | Sociotechnical barriers to decarbonization and their categorization.**
**a** Conceptual model of carbon lock-in as an iron triangle of reinforcing barriers: incumbent technology support systems, high transaction costs, and business risks of innovation. These points sustain carbon-intensive systems and impede low-

carbon transitions. **b** Classification of specific barriers of carbon lock-in into six categories: Cost, Fiscal, Regulatory, Statutory, Intellectual Property, and Other barriers. Adapted from Brown et al. [35].

The Intergovernmental Panel on Climate Change (IPCC) recommends decarbonizing construction sectors globally by 2050[12]. In April 2024, the United States Department of Energy (DOE) put forth a plan to decarbonize the building sector in the United States by 90% by 2050[13]. The 25 year strategy entails improving energy efficiency, electrifying buildings, shifting fuel usage to renewable sources, and increasing carbon sinks. To meet the 2050 goal, the construction sector must overcome its reluctance to adopt the low-carbon materials and processes and improve operational efficiency. Innovation in the building industry is particularly difficult, given its scale, its stepwise structure, and the vast network of stakeholders involved, which all increase costs of transition[14]. For firms, concerns about liability of unfamiliar technology, upfront costs, and market competitiveness coupled with unclear or outdated regulations have chilled widespread adoption of net-zero energy building (NZEB) technologies[15,16]. Globally, localized needs have made the building sector one of the "least amenable" to international cooperation, stalling coordinated action across borders[17].

Behaviorally, stakeholders are resistant to changing practices. This resistance is sustained by general lack of awareness of decarbonization and the necessity of updating practices[18,19], lack of data on technology performance[20,21], a limited number of professionals with the experience and technical expertise to carry out a NZEB[22–24], and lack of opportunities for ongoing professional training to incubate NZEB technology advocates[25–27]. Clients and investors are reluctant to adopt unconventional technologies because they lack awareness of benefits and tradeoffs and are reluctant to take on financial risk[28]. This uncertainty is fostered within a context of broader political ambiguity in the form of shifting subsidies or inconsistent policy targets. This ambiguity undermines confidence in the feasibility of long-term investments[29,30]. Top-down regulation is further fettered by misinformation, including exaggerated costs or doubts about the reliability of low-carbon technologies[31]. From a market standpoint, renewable energy infrastructure has generally been less profitable and more price-volatile than fossil fuels, which has depressed investments in wider renewable energy infrastructure[32].

Conceptually, the slow rate of change in the construction sector can be explained in terms of a multi-level "lock-in" framework. Institutional practices have become "locked-in," and resistant to change, by a combination of existing infrastructure, institutional practices, incentives, and ideologies that reinforce incumbent technologies[33]. This includes the continued reliance on fossil fuels in the building sector, despite clear long-term cost benefits of decarbonization technologies[34]. Brown et al.'s framework of "carbon lock-in" integrates social, political barriers with the well-recognized techno-economic

barriers to decarbonization[35]. This extended conceptual framework is useful given the construction sectors' multi-layered sectoral web of political, economic, institutional, and behavioral barriers. Carbon lock-in occurs as a result of the mutually reinforcing nature of the barriers and multipliers across building cycle phases[36]. Thus, the challenge to confronting barriers lies in disentangling the social structures that "lock" conventional practices in place. These barriers can be categorized within an "iron triangle" of costs, risks, and infrastructure (**see** Fig. 1a). Within this triangle, sub-barriers include: (1) high up-front costs, (2) lack of market incentives, (3) lack of clear policy frameworks, (4) unfavorable institutions and regulatory frameworks, and (5) incompatibility of existing infrastructure (see Fig. 1b)[35]. Carbon lock-in is informed by a sociotechnical framework that defines barriers as part of a complex, interdependent system, rather than as isolated blockades.

For many years, technology shortfalls were considered the critical impediment to decarbonization in the building sector[37–39]. However, in this study, by analyzing results of a systematic literature review, we show how overlapping social, economic, behavioral, and political barriers impede adoption of existing technologies of decarbonization. Perspectives of various stakeholders define the "optimal" pathways to decarbonization, which are often defined socially and politically, rather than by the best technical option[40]. The goal of this work is to provide an overview of the current research on sociotechnical barriers to decarbonizing the building sector.

The contributions of this study are empirical, conceptual, and methodological. First, the use of an interdisciplinary lens highlights social and sociotechnical barriers of both residential and non-residential buildings globally, including differences based on building stage, technology used, and location. Additionally, this work analyzes how barriers intersect to validate the utility of the carbon-lock-in framework. This study also adds an additional corpus of evidence from media and political testimony, extending the work examined in a traditional systematic review. Finally, the methodological approach extends the application of machine-learning to systematized literature reviews. These results can inform future directions for research and inform wider policy recommendations.

## Results
### Cataloging 95 sociotechnical barriers
The 95 barriers identified by our analysis were first grouped by analytically distinct barrier themes: economic, political, social/organizational, behavioral and technical (see Table 1). Economic barriers are the most prominent across the literature and include high upfront costs, lack of financing, additional costs of subcontractor training, updating

**Table 1 | List of 95 sociotechnical barriers to building decarbonization by theme**

| Economic | Political | Social | Behavioral | Technical |
|---|---|---|---|---|
| 1. High Costs | 26. Lack of Regulation | 50. Lack of Awareness of Decarb. Strategies | 72. Attitudinal Barrier | 83. Lack of Land for Growing Biomaterials |
| 2. Lack of Market Demand | 27. Reporting Requirements | 51. Lack of Stakeholder Alignment | 73. Developer Resistance | 84. Lack of Translatable Case Studies |
| 3. Long Return on Investment | 28. Lack of Government Support | 52. Lack of Management Support | 74. Health and Safety Concerns | 85. Lack of Understanding |
| 4. High Initial Investment | 29. Complexity of Assessment Tools | 53. Lack of Technical Training | 75. Aesthetic Concern | 86. Infrastructure Limitations |
| 5. Lack of Financing | 30. Lack of Policy Implementation | 54. Lack of Professional Training | 76. Lack of Client Interest | 87. Lack of Quality Substitution |
| 6. Higher Cost of Green Materials | 31. Lack of Auditing | 55. Lack of Accepted Standards | 77. Misinformation | 88. Poor Quality Existing BuildingSstock |
| 7. Higher Cost of Green Projects | 32. Lack of Compliance | 56. Lack of Project Coordination | 78. Stakeholder Resistance | 89. Lack of Local Data |
| 8. Lack of Incentive General | 33. Lack of Legislative Frameworks | 57. Lack of Skilled Professionals | 79. Client Resistance | 90. Lack of Space for Equipment |
| 9. Limited Green Suppliers | 34. Materials not Certified | 58. Lack of Technical Knowledge | 80. General Resistance to Change | 91. Lack of Supply Chain Capacity |
| 10. Uncertain Return on Investment | 35. Disconnect Local and Central Government | 59. Lack of Systems-Thinking | 81. Resistant Industry | 92. Technological Barrier General |
| 11. Time-Delay barriers | 36. Lack of Green Certification Systems | 60. Cyber Security Concerns | 82. Risk Uncertainty New Tech Adopt | 93. Relative Quality Concerns |
| 12. High Maintenance Costs | 37. Lack of Legislative Support | 61. Prefer Demolition over Deconstruction | | 94. Lack of Data |
| 13. Infrastructure Lock-In | 38. Lack of Material Standards | 62. Delays in Decision-Making | | 95. Lack of Commercial Applications |
| 14. Split Incentive | 39. Intellectual Property Rights | 63. Institutional Structure | | |
| 15. Cost of Additional Design Considerations | 40. Lack of Building Codes | 64. Lack of Public Awareness | | |
| 16. Cost of Waste Disposal | 41. Lengthy Planning and Approval | 65. Lack of On-Site Standard Application | | |
| 17. Higher Skilled Labor Cost | 42. Policy Uncertainty | 66. Low Status of Reused Materials | | |
| 18. Lack of Incentives to Retrofit | 43. Governance Barrier | 67. Lack of Stakeholder Awareness | | |
| 19. High Cost of New Model | 44. Heterogeneity of Policy Implementation | 68. Lack of Product Differentiation | | |
| 20. High Cost of Material Testing for Reuse | 45. Inconsistent Policy Language | 69. Lack of Project Team Interest | | |
| 21. Labor Intensive Installation | 46. Lack of Regional Standards | 70. Conflict of Interests among Stakeholders | | |
| 22. Cost of Developing New Standards | 47. Lack of Goals | 71. Lack of Marketing of New Technologies | | |
| 23. High Cost of Training Staff | 48. Regulatory Uncertainty | | | |
| 24. Higher Insurance Cost | 49. Statutory Uncertainty | | | |
| 25. Price Volatility | | | | |

Barriers are categorized into five themes: Economic, Political, Social, Behavioral, and Technical.

material standards, material cost premiums for unconventional materials, and uncertainty of returns on investment (ROI). Developers risk loss if changing market conditions produce lower selling prices than anticipated. While high upfront costs are a key barrier to scaling energy-efficient buildings, premiums are highly context-dependent and vary by building type, client, climate, and local incentives[41]. For example, facilities like hospitals or multifamily housing face steeper cost premiums for high performance systems, though they may also benefit more from long-term savings.

Politically themed barriers are the second most prevalent and include: lack of regulation, uncertain governance, and inconsistencies in policy implementation and oversight. In terms of regulation, building codes are an important impetus to improve efficiency; and, despite not having a federal code for building efficiency, the United States Department of Energy recommends that states adopt the International Energy Conservation Code (IECC)[42]. In Gainesville, Florida, stricter building codes introduced in 2002 led to a significant drop in residential energy use; new buildings used 4.27% less electricity and 6.67% less natural gas compared to those built before the code change[43]. Yet, some states lack a mandatory residential building energy code, leaving cities to adopt and enforce standards. This lack of concerted codes coupled with insufficient data on buildings' energy performance makes it difficult to assess which buildings are noncompliant and make systematic change[44,45]. Inflexibility in existing codes can further impede innovation and the adoption of unconventional low-carbon building materials. For example, some regions have been sluggish to adopt codes that allow the substitution of engineered wood product (i.e., Mass Timber) for steel in buildings above six stories, citing seismic concerns[46].

Thirdly, social barriers include a wider culture of resistance to innovation, lack of knowledgeable and skilled managers to advocate new technology, concerns about safety, and contractors' concerns about increased liability of adopting unfamiliar technologies. A lack of technical knowledge of, experience with, and advocacy for efficient technologies among contractors, tradespeople, auditors, and system analysts, has hindered technological changes[47,48]. A lack of technical knowledge, especially amid increasingly complex designs, can lead to stakeholder misalignment and cause delays due to rework, disputes, and contract changes[49]. Furthermore, poor site management can delay project completion and disrupt the construction schedule for green and conventional buildings alike[50].

The behavioral theme, though less discussed in the literature, points to barriers like a general lack of awareness about decarbonization practices and a strong risk aversion to unconventional materials or practices[51,52]. Risk aversion stems from homebuilders' exposure to warranty and defect claims, which can incur extra costs when using unfamiliar technologies. Even with strict design specifications, subcontractors may not always meet quality standards, leading to repairs under warranty that can be both financially burdensome and damaging to a builder's reputation.

Finally, the technical theme speaks to physical barriers such as: lack of infrastructure, lack of performance data, and lack of commercially available options or material substitutions (See Fig. 2). For example, adhesive-bonded wood products, though not new, have improved over the past 30 years, enabling the construction of high-rise wooden structures, acting as a steel alternative in some projects[46]. Still, some carbon-intensive materials, such as Portland cement, have been difficult to find adequate and feasible substitutions. Lower-carbon cement alternatives come with tradeoffs including longer setting time, less durability, and unresolved toxicity concerns[53].

## Spatial heterogeneity and geography
As for the geographic distribution of the literature in this sample, the bulk of publications are European cases (34%), followed by the United States (17%), China (16%), Australia and New Zealand (12%), and African

countries, (particularly Nigeria and South Africa) (8%) (See Fig. 3). The findings show a gap in the research on decarbonization technologies for low-income countries, which limits what comparisons can be made between mature and emerging markets. While all countries ranked economic barriers (i.e., high costs) among their top barriers, the United States' discussion housed a higher proportion of economic barriers in their top-ranking barriers than other countries/regions. While this particular market-emphasis shines in the United States, high upfront costs are consistent barriers across geographies. Additionally, all regions face a widespread lack of awareness of decarbonization practices and technologies.

Lack of technical knowledge and skilled workforce is particularly emphasized in the African research cases. This may be partially due to structural production dependencies; a "green division of labor" leaves Europe, North America, and parts of East Asia dominating patents and renewable energy manufacturing, making it difficult for African producers to compete or build capacity[54].

In China, political barriers are emphasized in the top barriers. Decarbonization of the building faces ongoing reliance on fossil fuel subsidies, complex administrative permitting and procedures, and a shortage of qualified assessors. These barriers persist despite national efforts to promote training and provide provincial incentives[55].

The European literature emphasized technical barriers above other regions including the difficulty in certifying salvaged materials for reuse, and insufficient performance data to address existing building performance.

Regional variations may indicate different expectations of who is responsible for managing the decarbonization transition (private or public sector), or may point to structural differences in state capacities, political economic distribution of decision-making, or human capital.

## Temporality
Publications of the barriers to decarbonization of buildings have increased over the last 15 years. In this time, there has been a small but significant decrease in discussions that focus explicitly on the economic barriers, and an increase in discussions of political, social, and behavioral barriers to decarbonization (see Fig. 4a). This points to a growing consensus in the academic literature on the intersectional nature of the barriers and the understanding of the importance of factors that extend beyond the traditional techno-economic framework. Still, while a one-way ANOVA test shows the increase in social and behavioral emphasis over time is significant, $(F_{(4,25)} = 4.03, p = 0.011)$, the most prominent barriers remain economic and political over time. In 2009–2011, the top three barriers were inconsistent policy language, lack of public awareness, and high costs. For 2012–2014, the top three barriers are high costs of green projects, high initial investment, and lack of building codes. For 2015–2017, the top barriers are high costs, time delays related to new technology, and lack of coordination across stages and stakeholders. For 2018–2021, top barriers are high costs of green materials, high initial investment, and time delays. Finally, for 2022–2024, high costs, high initial investment, and premium costs of green materials dominate. Thus, while specific challenges shift slightly over time, economic and political barriers have consistently remained the primary obstacles to decarbonizing the building sector.

## Building archetypes
By building type, residential buildings are the most prevalent within the discussion of barriers (44%), followed by commercial buildings (32%), educational buildings (10%), hotels (5%), hospitals (3%), other public buildings (2%), and data centers (1%) (Fig. 4b). The primary barriers to decarbonizing residential buildings are high costs, lack of incentives, and lack of building code requirements. Thus, economic, and regulatory challenges are most prominent in residential buildings.

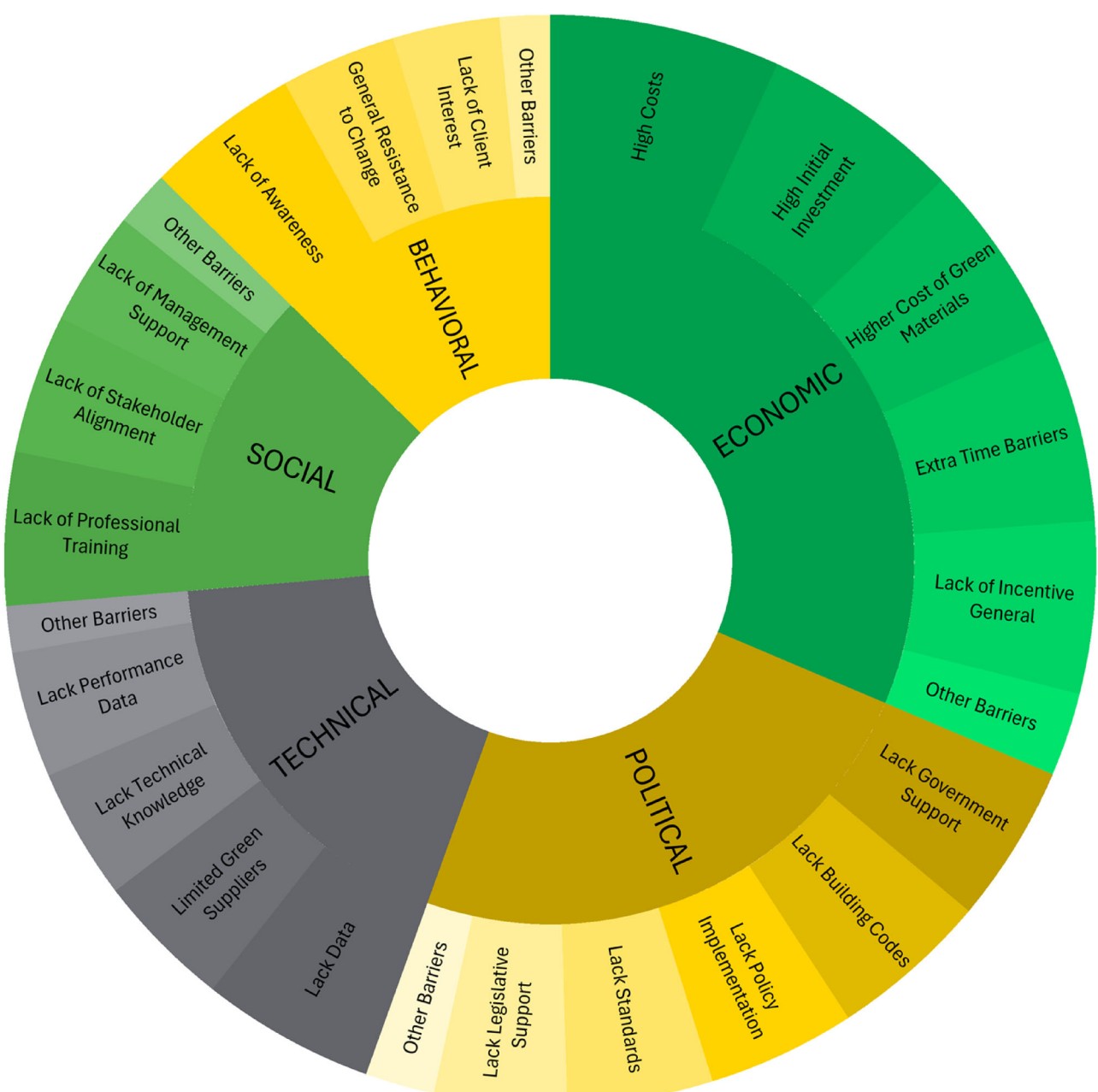

**Fig. 2 | Sociotechnical barriers to building decarbonization by theme.** Sunburst diagram displaying top sociotechnical barriers to decarbonization grouped into five themes: Economic, Political, Technical, Social/Organizational, and Behavioral. The inner ring represents the five themes; the outer ring shows key individual barriers within each. The size of each outer segment is proportional to its frequency in the academic literature, relative to the total dataset.

The top barriers for commercial buildings also include lack of standardization, specifically lacking agreed upon green certification systems, and lacking data to make decisions that optimize performance and cost-savings. Commercial buildings' top barriers indicate the additional barriers that might be overcome with better standardization of building requirements, more data availability on technology effectiveness, and incentives to support investments in low carbon technologies.

**Building lifecycle stages**

Buildings are understood through a phased life cycle because their greenhouse gas emissions and energy use differ across the phases of manufacturing, operation, and decommissioning (Fig. 4c)[56]. Barriers also vary by lifecycle phase, with the most concentrated discussion of

barriers clustering around standard setting and lack of designated goals for operation. Over time, literature increasingly highlights the growing importance of clear and consistent oversight, as uncertainty around regulations, standards, and political support can discourage upfront capital investment. Still, economic barriers are more prominent during building phases of material selection, construction, operation, and use. In decommissioning, economic disincentives are central, as the low cost of landfill disposal of construction waste and additional labor required for disassembly has supported an ongoing industry preference for demolition over deconstruction and made it difficult and costly to reuse building materials. Additional barriers to deconstruction include lack of demand for recycled materials, and the related lack of certification options for deconstructed materials for quality assurance.

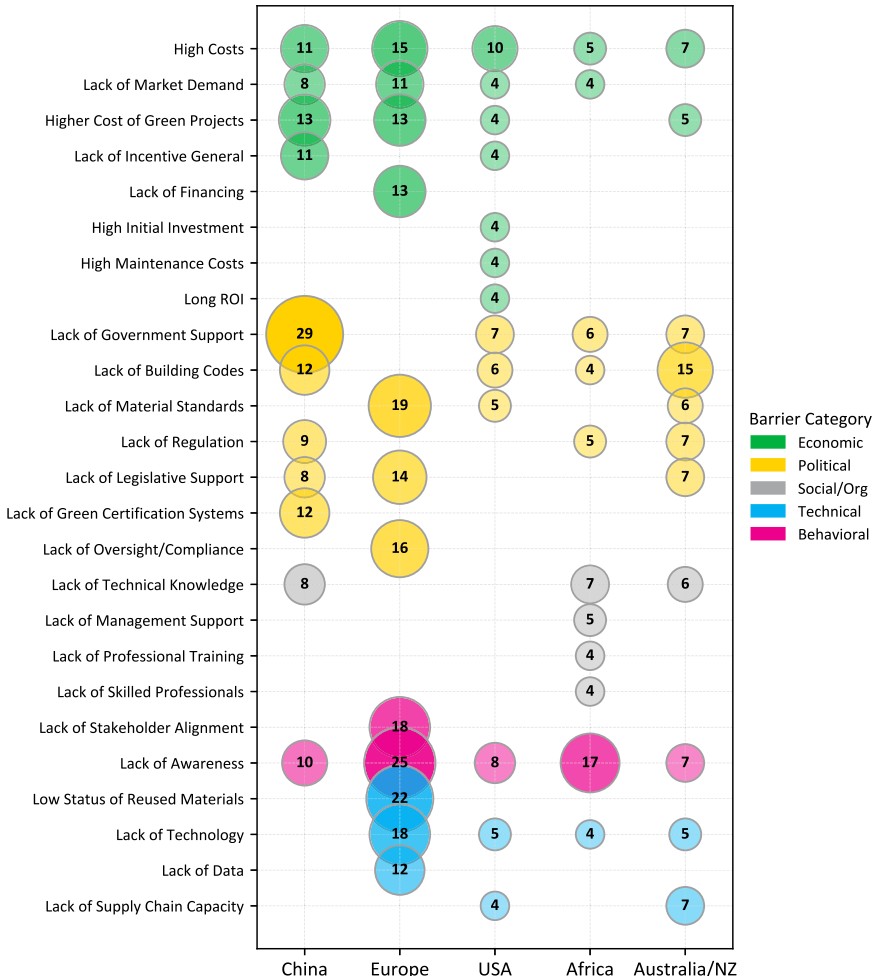

**Fig. 3 | Regional barriers to green building adoption.** Matrix dot plot showing the frequency of reported barriers across for cases discussed specifically for China, Europe, USA, Africa, and Australia/New Zealand. Circle size and color shade represent frequency; numbers indicate exact counts. Barrier categories are color-coded: Economic (green), Political (yellow), Social/Organizational (blue), Behavioral (pink) and Technical (gray). Empty cells indicate no responses for that barrier-region pair. *ROI* Return on Investment.

## Technological innovation and climate intervention types

Barriers also vary by decarbonization strategy. Here, the authors inductively draw from this data to classify technologies and other interventions into a nine-part framework (Fig. 5). Technologies in the decarbonization barriers literature are most often discussed in relation to developing green materials and adopting renewable energy. For example, a lack of supply chain for reclaimed materials means individuals have to search for and negotiate for materials and recertify them before reuse; the supply of reclaimed materials is further stunted by an industry-wide preference for building demolition over deconstruction[57,58]. Key material challenges include adapting supply chains and ensuring access to high-quality low-carbon materials. For one, building science is advancing on the performance of natural building materials, but these materials are not widely commercially available such as oil palm fiber based insulation[59]. Furthermore, bio-based insulation materials face several challenges that can limit their practical use without careful processing, including high moisture absorption, susceptibility to mold and decay, biodegradability, and high flammability[60]. Unsurprisingly, if new technologies, like biomaterials, cannot be easily integrated with conventional construction processes and methods, they are less likely to be adopted[61,62].

## Stakeholder priorities and institutional alignment

Lack of stakeholder alignment is identified as a key social barrier throughout the literature. This misalignment may arise in part because stakeholders involved in different phases of the building lifecycle have different priorities and understandings of optimal processes. Thematic analyses further reveal significant differences in the barriers prioritized by different stakeholder group (F(6, 539) = 8.97, $p$ < 0.001) (see Fig. 6). Economic barriers are most central for stakeholders who have the highest financial investment in the buildings (building owners and occupants). Predictably, policymakers emphasized political barriers, and engineers tended to highlight technical barriers over other stakeholders.

## Discussion

The above section outlined results as sorted by key themes from the literature. Here, this discussion further examines variation of barriers depending on the source of evidence (across media, scholarship or policy documents), to further consider implications of the results and research gaps, and to recommend examples of policy and applied action to overcome these locked-in barriers.

### Comparing themes across scientific, media, and political corpora

Differences have been identified between the innovations recognized by practitioners and those documented in academic literature[56] suggesting that barriers may be perceived differently depending on whether they are discussed in media, research, or policy. To further explore this, the authors compared the top barriers between the

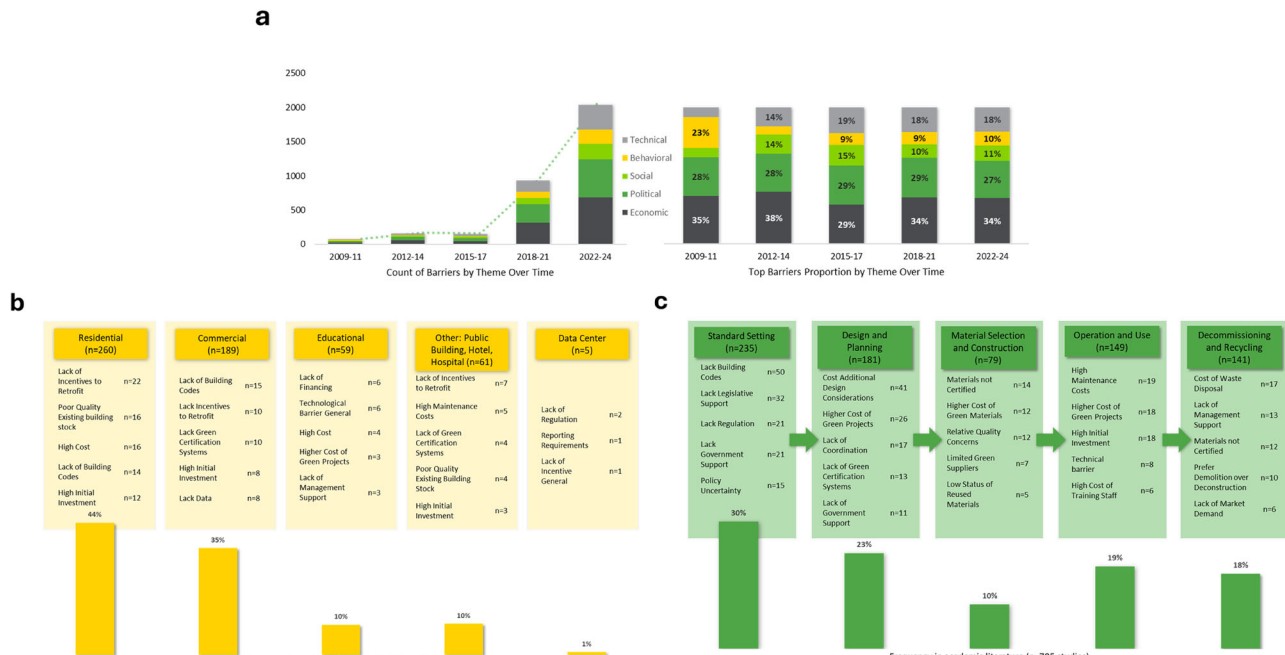

**Fig. 4 | The presence of sociotechnical barriers by time (a), building archetype (b), and lifecycle stage (c).** Panel 4a shows the count and proportion of socio-technical barriers to building decarbonization by theme in 3 year increments over 15 years (2009–2024). Panel 4a shows a sharp increase in discussion of barriers, especially after 2018. Economic and Political themes are most prevalent over this time. Colors represent barrier themes: Economic (dark gray), Political (medium green), Social (light green), Behavioral (yellow), and Technical (light gray). Fig. 4b: Frequency and top sociotechnical barriers to building decarbonization by sector across the 595 studies explicitly discussing sector. Residential and Commercial sectors are most prevalent in research literature. Fig. 4c shows the top socio-technical barriers to building decarbonization by building phase, based on 785 academic studies explicitly referencing phase. Barriers are grouped into five phases: Standard Setting (30%), Design and Planning (23%), Material Selection and Construction (10%), Operation and Use (19%), and Decommissioning and Recycling (18%). Each phase lists the most top reported barriers.

academic literature corpus, the dataset of political testimony and legislative documents, and the dataset of news media (Fig. 7). There are small, but significant, differences between the datasets (F(2,12) = 19.99, $p < 0.001$). Regardless of source, the most prevalent theme of barriers is economic, specifically, perceptions of high costs, and, when qualified, high upfront costs. Political sources stressed additional social and political barriers relative to the news or academic literature. Generally, the news media dataset discusses barriers with less frequency than political or academic sources, but when discussed, the barriers in news media tend to be economic barriers and are less likely to be behavioral barriers.

**Analytical strength of the carbon lock-in framework**
The barriers identified in this work validate the carbon-lock in framework and map on 65% of categories defined by Brown et al.'s carbon lock-in (see Table 2). The spaces without overlap show how carbon lock-in framework could be extended to account for the "other" categories including a stronger emphasis on contextual social/organizational and behavioral practices that also intersect to inhibit rapid decarbonization in the building sector. Specifically, the spaces lacking overlap highlight pathways to further research. For example, an area for future development is to investigate occupant behavior and energy use and its intersection with broader political, technical, and economic contexts[63].

However, not all of these barriers hold equal prominence. Economic barriers' prominence across the literature shows the significant role of upfront costs as a roadblock to transformative change within the building sector. In the view of stakeholders, decarbonization represents a cost which hampers the return on investment for construction projects. The stasis of economic barriers is further explained by their entanglement with social, political, and behavioral barriers, validating the carbon-lock-in framework. Still, the persistence of the prominence of economic barriers raises key questions on how such returns are calculated. Traditional methods of evaluating cost-benefit returns have been historically challenged[64]. In practice, accurate data of future costs and benefits are seldom integrated[65]; long-term additional expenses from failing to decarbonize buildings are generally overlooked, and co-benefits are generally ignored[66]. The choice of discount rate for decarbonization investments has also historically been a significant discussion point among economists[67,68]. As is, material substitution options can drive up pricing of green buildings[69], and trade restraints and material bottlenecks further exacerbate challenges of material availability and cost feasibility[70].

These challenges yield two critical considerations for industry. First, if industry can pass these additional costs of decarbonization on to tenants, it will overcome return on investment barriers for owners, but this is only true if all participants in the industry are treated fairly and equally. Adjacent to the cost conversation, the competitiveness of firms emerges as a central and complementary discussion, with key implications for policy. Second, the actual return on investment of the firm may not equate the overall social benefit of decarbonizing buildings, a well-known problem in economic theory.

The issue of distribution of costs naturally leads to political barriers. The political and economic context of the climate transition varies globally, defined by who is expected to lead change, with divergent governance models in China (state-led market), the United States (market-centric policy), and the EU (aligned state and market policy). In the United States, regulatory uncertainty and political inconsistency generate low confidence for long-term investments in green technology needed for building decarbonization[71,72]. Political barriers include lack of government and policy support, including legislative regulations, building codes, and standards that have the power to shape business-as-usual practices for the entire building sector. Institutions become entrenched in practices that help ensure

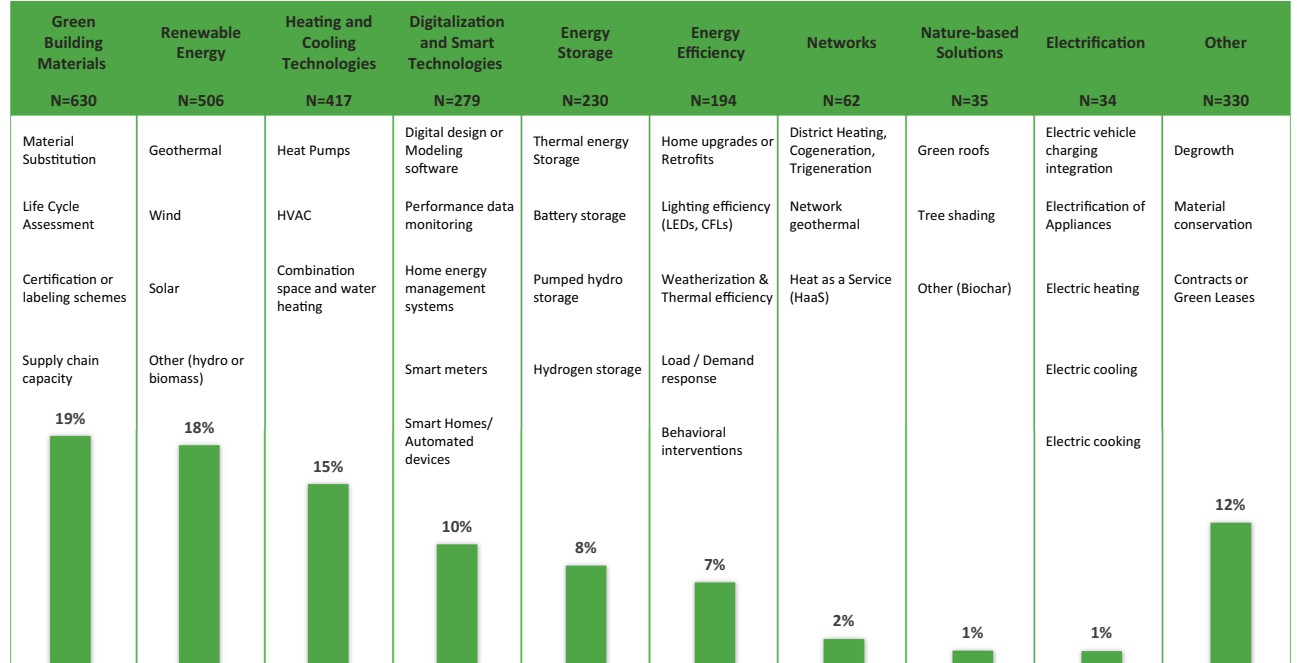

**Fig. 5 | Most prominent technologies and interventions discussed in the literature on decarbonization of buildings.** Decarbonization barriers are most frequently cited in discussions around Green Building Materials (19%) and Renewable Energy (18%), reflecting focused consideration of barriers around material substitution, certification schemes, and solar, wind, and geothermal technologies. These dominant categories housing barrier discussions contrast with sectors like Nature-based Solutions (1%).

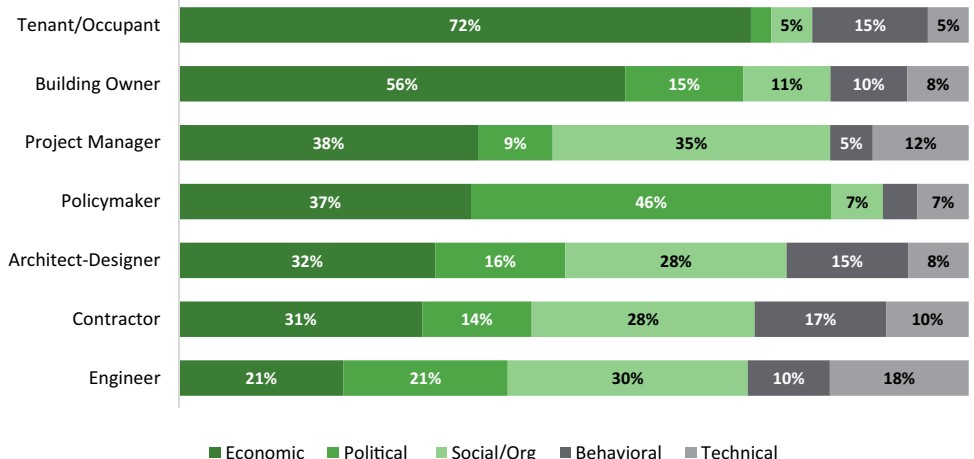

**Fig. 6 | Distribution of building decarbonization barrier types by stakeholder group.** Barriers are categorized into Economic (dark green), Political (medium green), Social/Organizational (light green), Behavioral (dark gray), and Technical (light gray) thematic categories and analyzed by stakeholder group and ranked by emphasis on economic barriers (*n* = 1190). Tenants and building owners most frequently report economic barriers. Policymakers emphasize political and economic barriers. Engineers emphasize more technical and social/organizational barriers.

their legitimacy and stability within an organizational field. Over time, firms within a sector become more homogenous, evolving in ways that minimize uncertainty, including adopting established practices, mimicking existing firms, and yielding to broader social conventions, such as standards and regulations[73,74]. These mechanisms to minimize financial risk, carry their own inertia, with a tendency towards stability, and resistance to disruptive change[75]. Known as "institutional isomorphism," this phenomenon is helps explain how lock-in operates at the meso-level and how overt explanations like perceptions of high cost, are interlaced with other social mechanisms, such as risk aversion. High costs offer basic explanation, but the question is why, even as decarbonization becomes economically feasible, adoption remains

sluggish. This work argues that carbon lock-in creates normative structures at all levels which have a strong inertia that resists change. The landscape of barriers provides some clues about these mechanisms. In this case, the secondary prominence of political barriers suggests that clear standards and regulations might be a powerful space to incentivize change and normalize high up-front investments in capital.

The second consideration of imbalanced investment to social benefit is more complex to apprehend. Outside of economic barriers which score highly across all stakeholders, the weight of other barriers varies significantly by role. Such variety demonstrates the misalignment across stakeholders on challenges related to decarbonization. At the institutional level, this misalignment is shaped by siloed processes

| Political Testimony and Legislation (n=175) | | | News Media (n=192) | | | Academic Sources (n=880) | | |
|---|---|---|---|---|---|---|---|---|
| High Costs | n=85 | 49% | High Costs | n=11 | 6% | High Costs | n=484 | 55% |
| Lack of Stakeholder Alignment | n=83 | 47% | Technological Barrier General | n=10 | 5% | High Initial Investment | n=425 | 48% |
| Lack of Professional/Technical Training | n=79 | 45% | Lack of Market Demand | n=10 | 5% | Higher Cost of Green Projects | n=403 | 46% |
| Technological Barrier General | n=77 | 44% | High Initial Investment | n=9 | 5% | Technological Barrier General | n=359 | 41% |
| Higher Skilled Labor Cost | n=70 | 40% | Lack of Stakeholder Alignment | n=8 | 4% | Lack of Market Demand | n=350 | 40% |
| High Initial Investment | n=70 | 40% | Lack of Regulation | n=8 | 4% | Lack of Building Codes | n=345 | 39% |
| Lack of Building Codes | n=55 | 31% | Lack of Policy Implementation | n=7 | 4% | Lack of Government Support | n=343 | 39% |
| Higher Cost of Green Projects | n=55 | 31% | Lack of Financing | n=7 | 4% | Lack of Awareness | n=328 | 37% |
| Lack of Awareness | n=51 | 29% | Lack of Government Support | n=6 | 3% | Time-related barriers | n=305 | 35% |
| Limited Green Suppliers | n=51 | 29% | Inconsistent Policy Language | n=6 | 3% | Lack of Data | n=299 | 34% |

**Distribution of Barrier Themes Within Source Group**

| Political Testimony and Legislation (n=175) | News Media (n=192) | Academic Sources (n=880) |
|---|---|---|

Legend: ■ Economic  ■ Political  ■ Social/Org  ■ Behavioral  ■ Technical

**Fig. 7 | Variation in building decarbonization barriers by political testimony, news media, and academic corpora.** Shows the distribution of sociotechnical barrier themes – Economic, Political, Social/Organizational, Behavioral, and Technical, across three source types: Political Testimony and Legislation, News Media, and Academic Sources. Each column displays the proportion of barrier types emphasized within each source group. Political (medium green) and economic barriers (dark green) more heavily weigh in political and media sources, while academic literature reflects a more balanced thematic spread, including greater attention to behavioral (yellow) and technical (light gray) barriers.

and different authorities with differing priorities and goals[76]. This finding is further reinforced by the distribution of social barriers across actors of the value chain (excluding tenants and building owners): social barriers are rooted in existing misalignment among stakeholders. Misalignment exists between the actors of the value chain, (for example, policymakers, building owners and tenants); and misalignment within the value chain, (as between engineers, contractors, architect-designers, project managers).

Misalignment on priorities and solutions impedes effective decarbonization of buildings while it complements and reinforces inertia of conventional practices. Part of the misalignment centers on the lack of professional training and management support. Organizational learning and knowledge transfer among industry actors also emerges a central roadblock for the decarbonization of buildings. Because inadequate procedures may hamper learning[77], knowledge transfer plays an important role, as both technology learning (technical barriers) and methods and processes (social barriers) are involved. In low-

or middle-income countries, the stated barriers suggest that investments might point to a more general need for technology transfer and skill building due to low institutional and infrastructure capacity[50]. It is crucial to spread new knowledge across the industry and foster coordinated change, either through direct implementation of new knowledge, as a source of inspiration for further innovation[75,78], or the development of new supply chains (technical barriers).

Misalignment in organizational decision-making processes can be rooted in corporate and individual behavioral barriers. These barriers notably include resistance or indifference to change (Fig. 2). Organizational culture can affect agency within an organization through critical lock-in patterns, such as cognitive bias, self-reinforcement bias, and confirmation bias[79]. Individual behavioral traits such as short-term focus, loss aversion, and the use of heuristics to simplify decision-making[80], can also reinforce inertia, leading to misalignment across different actors with varying goals, both within an organization and across organizations. These behavioral

**Table 2 | The barriers to building decarbonization according to the carbon lock-in framework**

| Carbon Lock-In Framework | Economic % | Political % | Social % | Behavioral % | Technical % |
|---|---|---|---|---|---|
| High Up-front Costs | 10 | 0 | 0 | 0 | 0 |
| Inadequate Technical Validation | 8 | 3 | 1 | 1 | 4 |
| Inadequate Workforce Competence | 3 | 0 | 2 | 0 | 4 |
| Uncertain Fiscal Policy | 3 | 1 | 0 | 0 | 0 |
| Inadequate Supply Channels | 1 | 1 | 2 | 0 | 1 |
| Volatile Energy Prices | 1 | 0 | 0 | 0 | 0 |
| Uncertain Regulations | 0 | 11 | 0 | 2 | 0 |
| Burdensome Permitting | 0 | 1 | 0 | 0 | 0 |
| Unfavorable Regulations | 0 | 0 | 1 | 0 | 0 |
| Infrastructure Limitations | 0 | 0 | 0 | 0 | 8 |
| Monopoly Power | 0 | 0 | 0 | 0 | 0 |
| Misinformation | 0 | 0 | 0 | 0 | 0 |
| Other | 11 | 7 | 8 | 8 | 1 |

Table of barrier thematic category and categories of previously established carbon lock-in concept. Columns indicate the percentage of barriers as clustered by Economic, Political, Social, Behavioral and Technical thematic categories, which align with 65% of the categories previously defined in carbon lock-in framework (see Fig. 1b, this paper). Economic barriers are most prominent, especially within infrastructure and institutional domains. Political, social, behavioral, and technical barriers are more unevenly represented. Gaps in coverage reveal opportunities to extend the framework and incorporate social/organizational and behavioral dynamics.

barriers further reinforce mimetic and normative pressures, contributing to institutional inertia.

Risk aversion, a significant behavioral barrier, is particularly prevalent in the heavily regulated construction industry, given health, safety, and liability concerns. This risk aversion becomes even more pronounced when considering new technologies at scale. Complex, large-scale innovative "megaprojects," typically costing over $1 billion, have a history of repeated failures[81,82]. For instance, the Sydney Opera House, an iconic megaproject, experienced a 1,400% budget overrun and took 10 years longer than initially planned to complete. Given its scale, the decarbonization transition of the construction sector is inherently risky, both financially and politically; thus, research is essential to continue to improve wider understanding of the inherent risks of new technology and institutions, because inaction and delaying decarbonization is already contributing to widespread health, economic, and environmental degradation[83,84].

The primary barrier to building decarbonization implementation centers on the challenge of upfront costs. Additional expenditures associated with decarbonization initiatives often fail to gain acceptance because they potentially compromise firms' competitive positions in the market. Consequently, the anticipated social benefits of decarbonization remain unrealized. This implementation barrier is exacerbated by multiple lock-in mechanisms operating at both the industry level (through normative and mimetic pressures) and organizational level (manifesting as risk aversion), which are further intensified by misalignment among diverse stakeholders. These challenges have significant implications for policy development. First, policymakers must prioritize establishing regulatory frameworks and institutional structures that foster a fair competitive environment among industry participants to drive systematic change. Second, they must focus on developing effective knowledge transfer mechanisms and information-sharing platforms to align dispersed stakeholders with divergent objectives. The focus should be on considering interventions across as many thematic areas as possible while continuously monitoring progress and making adjustments that align with the broader 2050 climate mitigation plan.

Since 2014, literature has increasingly focused on social and behavioral barriers to decarbonization, but more research is needed to identify the most critical areas where the building sector can adopt strategies to achieve 2050 carbon-neutral goals. Additionally, more building performance data is needed to pinpoint the key points where interventions could have the greatest impact and inform parsimonious

standards for building performance. As demonstrated in the carbon lock-in framework, barriers can overlap at the level of the individual, institutions and firms, and nation-states and as the trends over time suggest, addressing economic barriers alone are insufficient to capture the slow uptake of decarbonization. This persistent dominance of economic barriers is not separate from social factors that impact price, including lack of accountability for externalities cultural values, norms, and preferences. Lack of individual behavioral research is notable, including why there is a widespread barrier of lack of awareness of the process and benefits of decarbonization.

Another evident gap is that most studies, perhaps obviously, take a building or the building sector as its unit of analysis. But in doing so, the literature misses an opportunity to contextualize building decarbonization across its couplings to other systems, such as road infrastructure and traffic[85], or 'multi-systems' approaches that analyze buildings alongside other energy systems such as electricity supply, industry, mining, freight, and shipping[86].

Moreover, important social risks and energy justice concerns persist for the broader energy transition, including energy justice concerns for critical mineral mining and processing[87]. However, in the building-specific literature, there is limited discussion on environmental justice and social equity and related risks, and limited to material extraction and end-user cost[88]. A targeted review specifically on risks associated with building decarbonization would help to specify gaps and directions for essential research.

This work emphasizes the need to further investigate how a sociotechnical, integrated approach can explain the ways stakeholder strategies for decarbonization both shape and are shaped by broader social, political, and economic contexts. Future research should explore how different actors decide to adopt innovative technologies, which lock-in mechanisms matter most in specific contexts, and what drives early adopters especially when clear economic incentives are lacking. Additional research on decarbonization barriers in low-income countries can help create a comparative framework and inform passive design strategies and global standards to areas with low state capacity.

Despite these methodological constraints and gaps in the existing literature, particularly concerning low-income countries and emerging markets, this study highlights a complex landscape of barriers spanning the building lifecycle, stakeholder groups, publication sources, geographic contexts, and building types. At the heart of these challenges lies the fundamental issue of upfront costs, which emerges as a

**Building Codes**
- Mandatory compliance
- EU's Energy Performance of Buildings Directive: Zero emission by 2030
- International Energy Conservation Code: Energy Efficiency Performance Standard
- Phased Building Code, Boston

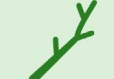

**Financial Incentives**
- Tax credits, subsidies, rebates, or penalties
- Supply Side: solar panel subsidies, heat pump subsidies, grid improvements
- Demand Side: rebates for efficient appliances, windows, retrofits

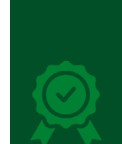

**Public-Private Partnerships**
- Contracted partnership to deliver policy programs
- Mass Save, Massachusetts: utility-state partnership for building assessment, workforce training, equipment rebates, installation

**Third-party Certifications**
- Voluntary performance labeling
- Third-party audits by International Standards Organization (ISO)
- LEED, Passive House, Living Building Challenge
- Building Operator Certification (BOC)

*Authority (Regulations)* · *Organization (Partnerships)* · *Treasure (Incentives)* · *Nodality (Information)*

**Fig. 8 | Visualizing policy recommendations according to the NATO/ATON framework.** This framework adapts Christopher Hood's "NATO" model to ATON, categorizing policy tools into four types: Authority (building codes and regulations), Treasure (financial incentives and market mechanisms), Organization (public-private partnerships and government programs), and Nodality (third-party certifications and informational tools). These policy tools guide compliance, investment, implementation, and accountability across both supply and demand in the building sector. Adapted from Hood and Margetts (2007)[89].

foundational barrier to implementing decarbonization initiatives in the building sector. These additional expenditures often face resistance because they potentially compromise firms' competitive positions in the market, leading to a critical gap between potential social benefits and realized outcomes.

The analysis demonstrates that implementation barriers are significantly exacerbated by multiple lock-in mechanisms operating at both industry and organizational levels. These manifest through normative and mimetic pressures within the industry, while risk aversion at the organizational level further compounds the challenges. The misalignment among diverse stakeholders intensifies these barriers, creating a complex web of resistance to change that spans technical, economic, and social dimensions.

The carbon lock-in framework strongly suggests that analysts should expand their focus beyond buildings and on-site technologies to address broader misalignment of technical, economic, political, and social conditions impeding acceptance of building decarbonization. This implies that efforts to decarbonize the buildings sector must not only respond to technical challenges but also create economic incentives, engender political support, and shape social and cultural attitudes through education and marketing. Furthermore, if one accepts that the interventions needed to decarbonize buildings are chosen for reasons other than the most optimal "technical" solution, then research and development pathways to promote low-carbon buildings must drastically change.

Based on this study, policy strategies are recommended to address prominent barriers to decarbonizing the building sector. Christopher Hood's "NATO" framework categorizes policy instruments into four types: Nodality (information and messaging), Authority (regulations), Treasure (financial incentives and market mechanisms), and Organization (government programs and agencies)[89,90]. While the sequence of implementation of decarbonization policies vary by country and have only a minor impact on $CO_2$ reduction (for example, reordering the elements of Hood's NATO acronym as "TOAN" or "ATON"), the most significant emissions reductions in mature, high-income markets like Central Europe or the United States come from the ability to set and enforce strict regulations[91]. In practice, the most effective policy packages integrate

approaches from across policy categories, with a strong emphasis on building regulation and enforcement capacity[92]. Thus, the authors reorganize Hood's categories and recommend an regulatory-forward policy approach (ATON) to address the locked-in nature of barriers to decarbonizing the building sector (see Fig. 8).

Authority instruments, the mandatory standards, laws, and building codes, serve as the regulatory "sticks" in policy toolkits. Stringent building codes can effectively enforce compliance and deter harmful practices; however, they also require administrative oversight to ensure compliance and incentives for implementation. Despite this, top-down regulations ensure all industry actors meet minimum performance benchmarks[93]. For example, the EU's Energy Performance of Buildings Directive mandates zero-emission for new buildings and major renovations by 2030[94]. Yet, inconsistent enforcement across localities demonstrates the need for capacity-building such as third-party audits and mandatory performance disclosure[95]. One example of a way to ramp up regulatory stringency is modeled in Boston, Massachusetts' three-part building energy code framework. Boston's tiered approach begins with a required Base Code which is aligned with the International Energy Conservation Code (IECC 2021) standards, followed by more rigorous opt-in Stretch Code emphasizing performance-based energy efficiency, and the final tier is an aspirational Specialized Code targeting net-zero emissions by 2050 through electrification[96]. The tiered structure encourages progression toward decarbonization by offering required base and voluntary "stretch" options which can be adopted according to municipality. Though regulatory policies can have regressive social impacts (disproportionately burdening low-income households), setting clear performance benchmarks is key to driving change, especially when paired with revenue-generating or compensatory policies.

Treasure instruments in the form of financial "carrots," incentivize efficiency, electrification, and renewable infrastructure through tax credits, subsidies, rebates, or penalties. Financial incentives support both supply-side solutions (e.g., solar panel subsidies, heat pump subsidies, grid infrastructure improvements) and demand-side actions (e.g., rebates for efficient appliances, windows, or energy retrofits). These instruments are especially important when new technologies face high upfront costs compared to incumbent technology.

To realize both regulation and its financial support apparatus, organizational instruments are needed in the form of on-the-ground initiatives supported through private-public partnerships[90]. Direct government provision of energy services affords government more control over the entire policy process, from design to implementation. In practice, financial instruments are often integrated with organizational instruments. For example, Mass Save, in Boston, is a utility-funded, state energy efficiency program that includes workforce training, building performance assessments, heat pump rebates, and other services for improving energy efficiency performance of new and existing buildings. Funded by fees from customer gas and electricity bills, Mass Save installed 40,000 heat pumps and weatherized 55,000 homes in Massachusetts in 2024[97]. This organizational instrument, while costly (~$1.3 billion a year), provides the incentive structure to support broader regulatory compliance and economic feasibility.

In contrast to these resource-intensive approaches, nodality instruments, such as energy performance labeling and voluntary certifications, are typically low-cost. While they can be effective in helping set goals and standardizing complex concepts like "social sustainability,"[98] their voluntary nature limits their enforcement power and impact. Still, nodality instruments play a vital role in fostering international collaboration, building technical expertise, and establishing common practices and standards. For example, the United States Building Operator Certification (BOC) provides training and a professional certification for energy-efficient operations and maintenance. This voluntary certification, launched in 1997 and now active in 37 United States' states and Canada, helps incentivize technical development by cultivating competitive market value of skilled workers who contribute to reduced energy use, improved building performance, and operational cost savings and further advance workforce development in support of decarbonization[99]. Voluntary professional credentials for LEED (Leadership in Energy and Environmental Design) and Passive House have been effective platforms to build a common language and standard of advanced building performance that have been leveraged for advocacy and integration into formal building codes. Nodality instruments like adopting voluntary certifications and credentials can serve to support community-building around shared professional resources and best practices, particularly in areas where authority instruments are ineffective or absent[100]. Indeed, LEED has played a central role in shaping the market value of green buildings. LEED-certified properties command an average 7% higher rental rate and sell at premiums ranging from 18–25% compared to conventional buildings. However, this premium tends to diminish as local building codes catch-up to the design and performance standards of LEED requisites[100], suggesting that part of the added value may stem more from a "novelty" premium than a purely "green" one[101]. Still, third-party certifications like LEED offer tangible financial advantages; for example, certified buildings have been shown to carry a 34% lower risk of loan default compared to their non-certified counterparts[102]. Thus, despite their limitations, nodality instruments can help drive market change and complement regulations to advance building sector decarbonization.

Beyond professional development, nodality instruments can also address behavioral barriers, such as the widespread risk aversion in the construction industry. Builders and subcontractors are hesitant to adopt unfamiliar technologies due to fear of warranty breaches or litigation if performance expectations are not met. And yet, this liability might be mitigated through third-party verification, such as audits by the International Standards Organization (ISO), which help ensure compliance with codes and validate performance claims[103]. Improving accuracy of energy modeling and following metrics of certification can help align stakeholder expectations, minimize greenwashing, and clarify a building's sustainability and efficiency profile. Further standardization, such as material certification and procurement guidelines, might further address performance uncertainties and even support a platform to certify salvaged material for reuse[104].

In addition, our paper captures barriers at only one point in time. The barriers have a temporality, in that their salience is both relative and dynamic: relative to a particular point in time and perhaps even space, and dynamic in terms of evolving or coevolving with other barriers. Indeed, some barriers may be coupled together, in that addressing one could positively, or negatively, affect others. For example, as building decarbonization becomes more affordable (addressing high costs) there may be knock-on effects for other barriers (such as social acceptance or the practices of building managers). Attempts to actively address these barriers in policy, regulation, and practice will need to become similarly holistic and dynamic as well, appreciating the whole-systems nature of the 95 barriers identified, and how they will evolve and change over time.

Given the evolving nature of technologies and market conditions, policy interventions must be adaptable and continually informed by building performance data and contextualized case studies to better account for the complex social and environmental impacts from material and process substitution throughout the building lifecycle. An effective policy approach to building decarbonization involves combining regulatory authority with financial incentives, organizational support, and informational tools, following an ATON policy structure. While the relative emphasis on each instrument may vary by context, integrating these tools can help address a range of structural, economic, and behavioral barriers and support a more coordinated and adaptable path toward emissions reduction in the building sector.

## Methods
### Critical systematized review
To map the existing barriers, this work relies on methodology drawn from three types of systematic literature reviews including a mapping review, a systematized review, and a critical review[105]. A mapping review allows researchers to identify research gaps and provides a descriptive overview of the literature to inform future research directions[105]. A systematized review is used to ensure transparency and reproducibility while synthesizing across studies with differing methodologies using inclusion and exclusion criteria and a reproducible Boolean search vocabulary and strategy[105]. Finally, as a critical review, this work includes an analysis of variations in the findings with the aim to provide conceptual innovation[105]. With the assistance of machine learning, the authors were able to extend the comprehensiveness of this work, reduce bias, and support a more comprehensive codebook for analysis.

### Search strategy
The research question asks what are the social, economic, political, and behavioral barriers to decarbonization of the building sector? To address this, the search used a Boolean syntax to include a vocabulary of keywords including:
1) Synonyms for decarbonization and subcategories (energy efficiency, electrification carbon sink, fuel transition)
2) Synonyms for buildings (residential, commercial)
3) Synonyms for barriers (challenges, impediments, issues)
4) Keywords to capture sociotechnical barriers (resistance, policy, and others).

The vocabulary syntax was used to search four databases for comprehensiveness including Environmental Complete (EBSCO), Web of Science: Core Collection, ProQuest, and Nexus Uni[106,107]. Using a pilot sample to test for relevance, the search vocabulary was then refined based on specific keywords to generate the final sample corpus (see Table S1 in supplemental information). The search, conducted in September 2024 generated a filtered sample of 3927 articles, that were extracted into Covidence software, then screened for relevance and

duplication by title and abstract for inclusion in the final sample which contains 880 academic resources, with additional 175 resources from United States' legislative documentation and political testimonials, and 192 resources from global news media. The authors completed an intercoder reliability assessment for relevance (>95%)[108].

### Inclusion and exclusion criteria

The literature review includes academic and trade documents from the last 15 years, starting in 2009, which capture the contemporary usage of the term decarbonization, which gains traction in usage in academic literature and media around 2011[109]. Given the time-sensitive nature of news and politics, the corpus of these materials contains results from the five previous years (September 2019–2024) (see Table S2 in Supplemental Information). Resources were included if they focused on buildings and the socio-technical barriers to decarbonization. Studies include qualitative and quantitative research (including literature reviews, interviews, and survey methodologies). Resources in this sample were published in English and address a subset of decarbonization as defined by the DOE: (1) energy efficiency, (2) electrification, (3) shift to low-carbon fuel sources, (4) increasing carbon capture and the barriers to their adoption in the following types of buildings: educational, office, hospital, retail, hotel or residential[16].

### Human coding triangulated with machine learning analysis

The codebook was developed from a process of deductive and inductive qualitative coding of key themes common in semi-grounded qualitative research[110]. Deductive coding was shaped by the guiding framework of carbon lock-in and the four-part research question. Manual coding, which requires reading line-by-line, interpreting content, and systematically identifying themes in the documents, generated 116 barriers to decarbonization, which were later condensed to ninety-five based on analytical similarity, and classified under five thematic categories guided by the initial research question (what are economic, political, social-organizational, behavioral, and technical barriers to decarbonization of the building sector). Manual qualitative coding of 10% of the resources was used to "train" NVivo's machine learning algorithm and generate results from the software's pattern-based autocoding feature[111]. (Examples of qualitative codes in Table S3). NVivo's pattern-based autocoding function relies on identification of similar codes to calculate statistical "term-frequency inverse-document frequency."[112] Human-guided pattern-based autocoding is a leading-edge approach to literature reviews that substantially lessens the labor and resources needed to complete a comprehensive literature review[113]. Machine-assisted coding can also reduce bias and improve the reliability and transparency of qualitative coding[114].

### Classification into academic, political, and media corpuses

This literature review includes three types of evidence. Academic work includes a global corpus of case studies, primary research, and secondary reviews to capture the scholarship on barriers to decarbonization in the building sector. Additional inclusion of political testimony and legislation provides insight into spaces of political resistance in the United States context. The political resources were collected from CQ Congressional Testimonies accessed through Nexis Uni. News media adds the potential to assess timely public sentiments on the discussion of decarbonization. Newspapers included both local and major world publications accessed through Nexis Uni, which was selected for its comprehensive database sourcing from over three thousand global news outlets.

### Limitations

To assist with the analysis of the corpus, the authors mobilized the use of machine-based learning capabilities to provide a reliable scope of the data. Given the machine-learning's inability to contextualize key words used to code, machine-learning can generate invalid results based on unorthodox use of the keywords or what other authors have called "high inference" codes that require high level of interpretation from the human coder[113]. The authors minimized invalid results through the use of manually specifying vocabulary to guide the machine-learning to generate low-inference codes, reporting findings at the document level, and revising vocabulary for autocoded outputs that deviated >15% from the manual coding. The use of reductionist vocabulary to guide the machine learning aids in processing the large codebook and quantity of resources analyzed for this work. As a technique, pattern-based autocoding is useful for literature reviews using large amounts of data to generate patterns and trends as used here.

## Data availability

The data, being the list of sources generated and analyzed in this in this study, are provided in the Supplementary Information file.

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

## Acknowledgements

This research was supported by Schneider Electric's Sustainability Research Institute (https://www.se.com/ww/en/insights/sustainability/sustainability-research-institute/) through grant #55211331.

## Author contributions

CRediT: E.H.: Conceptualization, Methodology, Writing—Original Draft, Data Curation, Formal analysis; Writing—Review and Editing, B.K.S.: Conceptualization, Methodology, Writing—Original Draft, Writing—Review and Editing, Project administration; T.K.: Conceptualization, Writing—Review and Editing, Project administration, Funding acquisition, V.P.: Conceptualization, Writing—Review and Editing, Funding acquisition.

## Competing interests

The authors declare no competing interests.
