## [Transparent Peer Review file · Nature Communications]

Reviewing the 95 sociotechnical barriers to the decarbonization of buildings

Corresponding Author: Dr Erin Heinz

Version 0:

Reviewer comments:

Reviewer #1

(Remarks to the Author)

The paper addresses a highly relevant and timely topic—the sociotechnical barriers to decarbonizing the building sector. It provides a comprehensive review of barriers categorized into economic, political, social, behavioral, and technical dimensions. While the paper is well-structured and offers valuable insights, there are areas that require improvement to enhance its clarity, depth, and overall impact.

1. While the paper identifies several barriers, the discussion on each category (economic, political, social, behavioral, technical) lacks sufficient depth. Provide more detailed examples or case studies to illustrate how these barriers manifest in real-world scenarios.
2. With the barriers identified, the paper could benefit from specific recommendations for overcoming these sociotechnical barriers. Add sections to discuss possible recommendations for mitigating the barriers.
3. While the paper addresses global challenges, it does not sufficiently differentiate between barriers in developed vs developing countries. I suggest you include a comparative analysis of how barriers differ by region or economic context.
4. Emphasize the dynamic nature of carbon lock-in. Explain how barriers evolve over time and how interventions need to adapt accordingly.
5. The policy recommendations should be more specific and actionable. Instead of simply calling for "policy interventions," provide concrete examples of the types of policies that could be effective. If possible, consider the political and economic feasibility of different policy options.
6. Some sentences are overly complex or awkwardly phrased. Simplify language for clarity and readability.

Reviewer #2

(Remarks to the Author)

The authors investigated a very hot topic for the construction sector, but following some notes and suggestions to improve the comprehension, the readability and the scientific soundness of the work.

The introduction is clear to frame the scene but the first suggestion is to update the energy consumptions data reference with more recent data and in general at the really beginning, first paragraphs specify the reference years used.

Figure 1A and B being a revision from previous version of other authors, it should be better represented, at this size is hard to read and comprehend and you should consider to use the same colours to support reader to connect the comprehension of A and B otherwise my suggestion is to prepare two distinct figures with dedicated descriptions.

Table 2 formatting should be checked and uniformized to the other tables.

In Figure 2 there are some areas of the graphs without specifications of the barriers.

After Table 8, the graph has not Figure n. and this figure needs a better explanation because it does not refer to the same 100% sample so it could be misunderstood and it should be clarified in the text.

The conclusions should be addressed more clearly referring to the previous section of limitations, research gaps in order to clarify and state in a more consistent way the added value of the study.

Version 1:

Reviewer comments:

Reviewer #1

(Remarks to the Author)

Thanks to the authors for addressing my comments. I have no further comments.

Reviewer #2

(Remarks to the Author)

THE AUTHORS FOLLOWED BOTH REVIEWERS COMMENTS AND THEY ADDRESSED THEM, IMPROVING THE QUALITY AND THE SCIENTIFIC SOUNDNESS OF THE WORK,
IN THIS FORM I RECOMMEND THE MANUSCRIPT FOR PUBLICATION.

May 22, 2025

Dear Reviewers,

Thank you for the opportunity to revise our manuscript: The 95 sociotechnical barriers to the decarbonization of buildings: NCOMMS-25-08092. In response to both the helpful reviews as well as the editorial request to expand the discussion on policy impacts, we have added a substantial subsection on policy recommendations, including specific local and global examples to illustrate their application.

We are grateful to the two peer reviewers for their thoughtful and constructive feedback. We have considered all reviewer comments and incorporated almost all of them into the revised paper, which we believe have significantly improved the clarity and quality of the manuscript. Major revisions include:

- Adding examples and case studies throughout manuscript, (particularly in Sections 4.1-4.6)
- Drafting an entirely new section on policy recommendations (Section 5.3)
- Updating visualizations and charts to improve readability
- Revising the conclusion to improve manuscript continuity

In terms of specific point-by-point responses:

REVIEWER 1	
The paper addresses a highly relevant and timely topic—the sociotechnical barriers to decarbonizing the building sector. It provides a comprehensive review of barriers categorized into economic, political, social, behavioral, and technical dimensions. While the paper is well-structured and offers valuable insights, there are areas that require improvement to enhance its clarity, depth, and overall impact.	We thank the reviewer for recognizing the contribution of this review and discussion (No action needed.)
1. While the paper identifies several barriers, the discussion on each category (economic, political, social, behavioral, technical) lacks sufficient depth. Provide more detailed examples or case studies to illustrate how these barriers manifest in real-world scenarios.	This is a good suggestion. To address this, we have incorporated specific examples to round out the findings Sections 4.1 - 4.6 . For example, Section 4.1 was rewritten to integrate examples for each barrier thematic group (economic, political, social, behavioral, technical).
2. With the barriers identified, the paper could benefit from specific recommendations for overcoming these sociotechnical barriers. Add sections to discuss possible recommendations for mitigating the barriers.	We agree with the reviewer’s suggestion to add more specific policy recommendations to address these barriers to decarbonization and include local and global examples. To address this and thereby increase the applicability and contribution of the paper, we have added an additional substantive subsection to the final discussion. This Section 5.3 is organized by Hood’s “NATO” framework and begins:

	5.3 Recommended policy and applied action to overcome these locked-in barriers Based on this review, policy strategies are recommended to address prominent barriers to decarbonizing the building sector. Christopher Hood’s NATO framework categorizes policy instruments into four types: Nodality (information and messaging), Authority (regulations), Treasure (financial incentives and market mechanisms), and Organization (government programs and agencies)...
3. While the paper addresses global challenges, it does not sufficiently differentiate between barriers in developed vs developing countries. I suggest you include a comparative analysis of how barriers differ by region or economic context.	We thank the reviewer for this observation. We agree that this is a clear gap in the literature and one that should be addressed in future research. To clarify this gap in the literature, we add this statement in Section 4.2: “A limitation revealed in this review is the lack of research on decarbonization technologies for low-income countries, which limits what comparisons can be made between high and low-income countries.” We also specify this as an important future direction for research in Section 5.2: “Additional research on decarbonization barriers in low-income countries can help create a comparative framework and inform passive design strategies and global standards to areas with low state capacity...”
4. Emphasize the dynamic nature of carbon lock-in. Explain how barriers evolve over time and how interventions need to adapt accordingly.	We thank the reviewer for this insight. It is entirely correct, that our paper captures barriers at only one point in time. They do have a temporality to them in that their salience is both relative and dynamic: relative to a particular point in time and perhaps even space, and dynamic in terms of evolving or coevolving in lieu of other barriers. Some barriers may even be coupled together, in that addressing one could positively, or negatively, affect others, such as making building decarbonization more affordable (addressing high costs) having knock-on effects for other barriers (such as social acceptance or the practices of building managers). Attempts to actively address these barriers in policy, regulation, and practice will need to become similarly holistic and dynamic as well, appreciating the whole-systems nature of the 95

	barriers identified, as well as how they will evolve and change over time. We have now added language to this effect to the paper itself in Section 5.3. And, while we illuminate how barriers change over the 15 years of the sample (see Section 4.3) we agree with the reviewer that it would be fascinating to see how strategies and policies to address barriers evolve over time. This is a generative space for further research.
5. The policy recommendations should be more specific and actionable. Instead of simply calling for "policy interventions," provide concrete examples of the types of policies that could be effective. If possible, consider the political and economic feasibility of different policy options.	We integrate this suggestion with the above recommendation of the reviewer to offer more specific actions to overcome barriers. This is an important recommendation as we believe addressing this recommendation from the reviewer will extend the applicability and contribution of this work. To offer specific policies (which we see as effective in overcoming some of the most prominent barriers in the sector) we offer a new Section 5.3: Recommended policy and applied action to overcome these locked-in barriers, which offers policy directions and recommendations based on the findings of the report.
6. Some sentences are overly complex or awkwardly phrased. Simplify language for clarity and readability.	We thank the reviewer for sharing the experience of the reader. We have revised the document with the aim of reducing passive sentences (reduced by 2%), simplifying syntax, and shortening sentences to improve readability.
REVIEWER 2	
Reviewer #2 (Remarks to the Author): The authors investigated a very hot topic for the construction sector, but following some notes and suggestions to improve the comprehension, the readability and the scientific soundness of the work.	We thank the reviewer for recognizing the timeliness of the work. (No action needed)
The introduction is clear to frame the scene but the first suggestion is to update the energy consumptions data reference with more recent data and in general at the really beginning, first paragraphs specify the reference years used.	A very helpful comment and now addressed. In response we have updated the energy consumption reference [2,3] to reflect the most recent EPA data: EPA (2024) Inventory of U.S. Greenhouse Gas Emissions and Sinks: 1990-2022. U.S. Environmental Protection Agency, EPA 430-R-24-004. https://www.epa.gov/ghgemissions/inventory-

	us-greenhouse-gas-emissions-and-sinks-1990-2022.
Figure 1A and B being a revision from previous version of other authors, it should be better represented, at this size is hard to read and comprehend and you should consider to use the same colours to support reader to connect the comprehension of A and B otherwise my suggestion is to prepare two distinct figures with dedicated descriptions.	This is an important suggestion to improve readability, and we thank the reviewer for this feedback. We have re-sized the colors the panels in Figure 1 to be more accessible. We have also updated the colors to align conceptually across the figures.
Table 2 formatting should be checked and uniformized to the other tables.	We thank the reviewer for this careful attention to detail. We have now updated the colors and formatting of Table 2 to match the formatting of other tables in the manuscript.
In Figure 2 there are some areas of the graphs without specifications of the barriers.	We thank the reviewer for this careful attention to detail. The formatting for Figure 2 has now been adjusted (made a bit larger) to make sure that the labels are visible.
After Table 8, the graph has not Figure n. and this figure needs a better explanation because it does not refer to the same 100% sample so it could be misunderstood and it should be clarify in the text.	Again, this reviewer offers a very helpful suggestion to help with readability. We have updated the format of Table 8 to more clearly indicate how the chart refers to the data in the above table and added a subheading that reads “Distribution of Barrier Themes Within Source Group” to clarify the distinction of percentage present from the percentage distribution within the source groups.
The conclusions should be addressed more clearly referring to the previous section of limitations, research gaps in order to clarify and state in a more consistent way the added value of the study.	We thank the reviewer for this suggestion, as we also think this addition improves the continuity of the manuscript. To address this, we added language in Section 6. Conclusion: “ This study’s findings should be understood as high-level thematic insights, rather than deep interpretive analysis, due to the limitations of machine learning in capturing context-dependent or high-inference codes. Despite these methodological constraints and gaps in the existing literature, particularly concerning low-income countries and emerging markets, the research highlights a complex landscape of barriers spanning the building lifecycle, stakeholder groups, publication sources, geographic contexts, and building types...”